# HS-SPME Analysis of True Lavender (*Lavandula angustifolia* Mill.) Leaves Treated by Various Drying Methods

**DOI:** 10.3390/molecules24040764

**Published:** 2019-02-20

**Authors:** Jacek Łyczko, Klaudiusz Jałoszyński, Mariusz Surma, Klaudia Masztalerz, Antoni Szumny

**Affiliations:** 1Faculty of Biotechnology and Food Science, Wrocław University of Environmental and Life Sciences, Norwida 25, 50-375 Wrocław, Poland; antoni.szumny@upwr.edu.pl; 2Institute of Agricultural Engineering, Wrocław University of Environmental and Life Sciences, Chełmońskiego 37-41, 51-630 Wrocław, Poland; klaudiusz.jaloszynski@upwr.edu.pl (K.J.); mariusz.surma@upwr.edu.pl (M.S.); klaudia.urbanska@upwr.edu.pl (K.M.)

**Keywords:** essential oil, drying, SPME, true lavender, volatile constituents

## Abstract

True lavender (*Lavandula angustifolia* Mill.) is a widely used flavoring and medicinal plant, which strong aroma is mainly composed of linalool and linalyl acetate. The most valuable parts of the plant are the flowers, however leaves are also abundant in volatile constituents. One of the main factors responsible for its quality is the preservation procedure, which usually comes down to a drying process. For this reason an attempt to verify the influence of various drying methods (convective drying, vacuum-microwave drying and combined convection pre-drying with vacuum-microwave finishing drying) on the quality of true lavender leaves was carried out by determination of the volatile constituents profile by solid-phase microextraction (SPME) coupled with GC-MS technique. Total essential oil (EO) content was also verified. The study has revealed that the optimal drying method is strongly dependent on the purpose of the product. For flavoring properties convective drying at 60 °C is the most optimal method, while the best for preserving the highest amount of EO is vacuum-microwave drying at 480 W. Furthermore, SPME analysis had shown that drying may increase the value of true lavender leaves by significantly affecting the linalool to linalyl acetate to camphor ratio in the volatile profile.

## 1. Introduction

*Lavandula angustifolia* Mill. (also named *Lavandula officinalis* Chaix)—the true lavender—is a essential oil-bearing plant known worldwide, which history of usage starts in Greek and Roman times and last up to this day. The entire genus belongs to the large *Lamiacae* family, which is mostly native to the Mediterranean region, however true lavender is a commonly growing plant in England, Europe, North America and Australia. The most valuable part of the plant are flowers due to their much higher essential oil content than leaves, and a favorable linalool to linalyl aceate to camphor ratio [1]. 

Nowadays due to the well-recognizable aroma lavender plants or their derivatives find applications in numerous ways, like in perfumery, cosmetics and household products, antimicrobial agents, food fragrance and flavor improvement or as food preservatives [1,2,3]. Furthermore, the essential oil obtained from lavender is an interesting object for trials considering biological activity and even in medicinal trials. Some studies and overviews from recent years mention the anti-aging, analgesic, nuroprotective, sedative or anticancer activities of lavender essential oil [2,3,4,5,6,7,8,9]. These various lavender essential oil applications are due to their unique chemical composition, rich in monoterpenes, sesquiterpenes, sesquiterpenoids, aliphatic compounds and especially an abundance of monoterpenoids [10], with linalool and linalyl acetate highlighted as main flower components [1,11,12]. In the case of the leaves the main essential oil constituents are eucalyptol (1,8-cineole), camphor and borneol [13,14,15].

As the main factors affecting the quality of the essential oils obtained from essential oils-bearing plants, plant chemotype, growing conditions and location, fertilizers used, time of harvesting and post-harvest treatment (including preservation method) are mentioned [2,11]. Among those factors, the preservation method has the most significant influence, where the most common one for plants rich in essential oils is drying [16,17,18]. Drying of essential oil-bearing plants allows one to obtain sustainable products with guaranteed quality, although it may cause also considerable losses of valuable constituents—mainly affecting the volatile constituents [17]. Furthermore, the color of the raw material may be strongly influenced by drying [16].

The traditional and natural method of drying uses solar radiation, however nowadays convective drying (CD), which uses flows of the hot air [17,18], is the most common drying method used in natural products treatment. Nevertheless other techniques like freeze-drying, infrared drying, vacuum-microwave drying (VMD), spray drying or a combination of convective pre-drying with vacuum-microwave finishing drying (CPD-VMFD) are lately the objects of numerous investigations regarding natural products drying [18]. Unfortunately in case of drying the true lavender leaves only single factors were investigated. Interest in this topic is due to the necessity to find an optimal drying method for specific raw materials. In addition, not only a specific technique, but also its parameters, like drying time, temperature or pressure have a significant influence on the quality of the obtained products [19,20,21,22]. Overall the most important are air velocity and temperature—for plants the most suitable temperature is one between 50 °C and 60 °C [16]. 

The objective of this study was to determine the volatile profile composition and compound quantity of true lavender leaves and the influence of three drying methods (CD, VMD, CPD-VMFD) applied with various parameters. The study was done by a solid-phase microextraction (SPME) coupled with gas chromatography mass spectrometry technique (GC-MS). Also the total essential oil content was validated by using a hydrodistillation extraction technique.

## 2. Results and Discussion

### 2.1. Drying Kinetics

Figure 1 shows changes with time of the moisture ratio (MR) of leaf samples dehydrated by VMD at three magnetron powers (240, 360 and 480 W, Figure 1a), CD at temperatures in the range of 50 to 70 °C (Figure 1b), and combined (CPD-VMD) drying consisting of CD at 60 °C and VMD at a magnetron power of 480 W (Figure 1c). The drying times, together with the maximum temperatures, the final moisture content and the constants of the Page model are listed in Table 1. 

The Page model can be successfully used to describe the drying kinetics of the true lavender leaves dehydrated by the CD, VMD and CPD-VMD methods, characterized by high values of the determination coefficient (R^2^ > 0.99) and low RMSE values (<0.05). A good adaptation of the applied Page model for description of the drying kinetics can be found in many earlier publications of dill leaves, chanterelle and oyster mushrooms [23,24,25].

In the case of CD increasing the drying air temperature from 50 to 70 °C decreased the time of drying from 245 to 135 min, respectively. In VMD drying, radical reductions in the total drying time have been observed: the time was shortened from 32 to 14 min with a power change from 240 to 480 W. This radical reduction in the total drying time of VMD compared to CD is a result of the conventional water diffusion occurring, according to Fick’s law, that is supported by a pressure diffusion mechanism of the Darcy type [26]. Combined CPD and VMFD using 480 W, shortened the drying time of leaves almost 18-fold compared to CD at 50 °C. The use of CD and 480 W power caused a drop in the material temperature during VMD by 4 °C for leaves and 2 °C for flowers in reference to VMD 480 W. This condition is caused by the molecular distribution of water particles inside the dried CD and the distribution of water particles has an effect on the generation of heat energy production under microwave radiation during VMD [21,27,28]. Energy consumption during the CD of plant materials is much lower than in VMD [29,30]. In industrial conditions, the best solution is a combined drying process consisting of CPD and VMFD. The CD is very effective at the beginning of the drying process (the largest loss of water occurs during that phase) and VMD at the final stage of drying (removal of water strongly bound to the cellular structure of the material being dried) [18,27,28]. The final choice of recommended drying process should be related to the aspects of the dried material (volatile composition and sensory attributes) [27,31].

### 2.2. Volatile Constituents Profile of Fresh True Lavender Leaves

HS-SPME analysis coupled with the GC-MS technique had revealed one hundred and four peaks (one as a two compound mixture) recognized as volatile constituents, of which only one hundred of them could be identified (the mass spectra of unidentified constituents are available in Appendix A). Volatile constituents of true lavender leaves are listed in Table 2. Among them nineteen compounds were qualified as monoterpene hydrocarbons, twenty-six as oxygenated monoterpenes, twenty-four as sesquiterpene hydrocarbons, nine as oxygenated sesquiterpenes, ten as esters and eleven as others.

The main headspace volatile constituents of the examined true lavender leaves samples were *p*-cymen-8-ol (4.09% ± 0.67), a mixture of borneol and lavandulol (4.66% ± 0.69), *o*-cymene (4.81% ± 0.52), bornyl acetate (5.57% ± 0.82), (*E*)-caryophyllene (6.11% ± 1.48), eucalyptol (7.28% ± 1.06) and γ-cadinene (10.53 ± 1.51). In less amounts cumin aldehyde (1.92% ± 0.59), τ-cadinol (2.04% ± 0.55), *m*-cymen-8-ol (2.09% ± 0.25), camphor (2.09% ± 0.92), *p*-cymene (2.58% ± 0.33), caryophyllene oxide (3.31% ± 0.18), limonene (3.42% ± 1.16) and 1-octen-3-pl acetate (3.80% ± 0.52), which have a significant influence on true lavender leaves’ fragrance quality, were identified. The most characteristic and valuable constituents for true lavender (flowers), linalool and linalyl acetate, represented 0.42% ± 0.03 and 2.21% ± 0.73 of the total amount of volatile constituents, respectively.

Similar findings were reported in recent studies where eucalyptol (8.50% and 31.9%), borneol (15.21% and 24%), camphor (2.00% and 16.1%), cumin aldehyde (0.50% and 2.2%) were identified as main volatile components of a true lavender leaves sample [33,34]. Also, one of these studies, by Hassanpouraghdam et al. [34] pointed out low amounts or even a lack of linalool (0.7%) and linalyl acetate. This result is contrary to the one obtained in this study, however it may be related to the slightly different plant chemotype or due to the fact that in Hassanpouraghdam’s study leaves essential oil was analyzed, not headspace volatiles. Nurzyńska-Wierdak and Zawiślak [35] have identified linalool and linalyl acetate in a similar ratio (1:5), and furthermore they also found higher amounts of γ-cadinene (3.4 ± 0.1) and caryophyllene oxide (7.2% ± 0.2).

Unfortunately, there is a lack of reports in literature including HS-SPME analysis of true lavender leaves volatile constituents. Most of available ones takes as study object lavender flowers or whole aerial parts of the plant, where linalool and linalyl acetate dominate in the chromatographic profile of the volatile constituents [36,37,38]. Torabbeigi and Aberoomand Azar [39] reported high amounts of eucalyptol (41.37%), camphor (15.83%), borneol (12.32%), α-pinene (4.66%), and γ-cadinene (1.07%) found by HS-SPME analysis of true lavender samples. At the same time they did not find any traces of linalool or linalyl acetate, suggesting that the major part of their samples were lavender leaves.

### 2.3. Effect of the Drying Methods on the Quantity of True Lavender Leaves Volatile Constituents

In the fresh true lavender leaves cultivated in Poland used in this study the content of essential oil was 3.082 g per 100 g^−1^ of DW. Overall this essential oil yield is high in comparison to previously reported ones, as Mirahmadi and Norouzi [40] obtained just 2.34% of essential oil from true lavender. Moreover, Milojević et al. [41] report the essential oil yield in sage and eucalyptus leaves ranges from 2% up to 2.87%. Changes of essential oil content, the concentration of sixteen major volatile constituents and linalool caused by the various drying methods are shown in Table 3.

In the case of essential oil content all applied drying methods significantly affected the raw material. The most efficient method was VMD 480 W (1.302 g per 100 g^−1^), followed by VMD 240 W (1.075 g 100 g^−1^), CD 70 °C (0.992 g per 100 g^−1^) and CPD-VMFD (0.921 g per 100 g^−1^) which were in overlapping significant groups. The percent recovery of essential oil in these methods were as follows 42.26%, 34.87%, 32.19% and 29.87%, in comparison to the amount of essential oil obtained from fresh sample. The less efficient drying method was CD 50°C, with a 19.06% recovery. The ratios of percent recovery between fresh sample and ones subjected to drying are presented in Figure 2. Baydar and Erbaş [42], Figiel et al. [19], Ghasemi et al. [43] found as well that due to the applied drying method or its parameters the decrease in essential oil yield of green plant parts may range as high as three to five times. Furthermore, Politowicz et al. [24] and Nöfer et al. [27], in the case of mushroom drying, observed similar effects to the ones found in this study.

The total essential oil content results are not equivalent to the content of sixteen major constituents. In fresh true lavender leaves, sixteen major constituents accounting for 66.51% of the total volatile constituents and changes caused by all drying methods were significantly distinct. Less differences were observed in the CD and CPD-VMFD methods (5.09–6.68 percentage points) and the highest were observed for the VMD method (22.54–24.04 percentage points). Further, some results in the case of particular constituents among the sixteen major ones are worth underlining. Again, all drying methods had a significant influence on a particular constituent share of total volatile constituents. The most interesting was the increase a share of linalyl acetate (even up to 11.06% of the share in the CD 60 °C method) along with the decrease of camphor share (down to 1.40%) at the same time. Also the share of linalool, the main aroma compound for true lavender, increased significantly after all drying treatments, except for VMD 240 W and VMD 360 W. These results suggest that applying drying, mainly CD, for true lavender leaves, may improve the characteristics for use in flavoring, in accordance with Kim and Lee [44] and Da Porto and Decorti [45], who report that the high ratio of linalool and linalyl acetate to camphor ratio is an important quality marker for lavender fragrance. Similar changes after applying drying were obtained by Śmigielski et al. [10]. Nevertheless, if the aim is to preserve as much essential oil as possible, the VMD methods would be more applicable. Very poor results, both in case of total essential oil and major volatile constituents, were obtained after the CPD-VMFD method, what is in contradiction with results obtained by Szumny et al. [20] for rosemary drying *(R. officinalis*), however the taxonomic differences between rosemary and true lavender should be considered. 

## 3. Materials and Methods 

### 3.1. Plant Material

The drying process was carried out on true lavender cultivated in Poland (Kawon-Hurt Nowak Sp.j. Company, Gostyń, Poland). The initial moisture content of material was 2.7 kg per kg of dry weight. The drying processes were stopped after no further change in weights was observed. Moisture content of samples was determined using a vacuum dryer (SPT-200. ZEAMIL Horyzont, Krakow, Poland).

### 3.2. Drying Methods

#### 3.2.1. Convective Drying (CD)

CD was performed using the equipment designed and constructed at the Institute of Agricultural Engineering (Wrocław University of Environmental and Life Sciences, Wrocław, Poland). Samples were placed in the container (d = 100 mm) and dried at 50 °C. 60 °C and 70 °C—all with an air velocity of 0.5 ms^−1^. 

#### 3.2.2. Vacuum-Microwave Drying (VMD)

VMD was performed on samples with a SM 200 dryer (Plazmatronika, Wrocław, Poland). The dryer was equipped in cylindrical drum made of glass (18 cm of diameter × 27 cm of length). The drum with glass rotated with 6 rev·min^−1^. In the dryer system there was a BL 30P vacuum pump (Tepro, Koszalin, Poland), an MP 211 vacuum gauge (Elvac, Bobolice, Poland) and a compensation reservoir of 0.15 m^3^ capacity and a cylindrical tank. In this study, three power levels (240, 360 and 480 W) and pressures ranging from of 4 to 6 kPa were used. The maximum temperature of dried lavender leaves was measured right after the removal from the dryer using an i50 infrared camera (Flir Systems AB, Stockholm, Sweden).

#### 3.2.3. Combined Drying—Pre-Drying by Convective Drying with Vacuum-Microwave Finishing-Drying (CPD-VMFD)

CPD-VMFD performed on samples consisted of CPD at a temperature of 60 °C until a moisture content of leaves was around 0.45 kg·kg^−1^ db, was reached, followed by VMFD at 480 W.

### 3.3. Modeling of Drying Kinetics

The drying kinetics of CD, VMD and CPD-VMFD were fitted based on the mass losses of the true lavender samples. For CD, weight losses were monitored every 2 min for the initial 20 min and then every 5 min thereafter until the end of the drying process.

VMD samples were monitored every 2, 3 and 4 min for 480, 360 and 240 W. Different drying time intervals were applied in order to ensure a similar energy input between subsequent measurements regardless of the microwave power level.

The moisture ratio (MR) of lavender leaves during drying experiments was calculated using the following equation:(1)$$MR=\frac{M_{\left(t\right)}-M_{e}}{M_{o}-M_{e}}$$
where M_(t)_ is the moisture content at time τ. M_o_ is the initial moisture content, and M_e_ is the equilibrium moisture content (kg water/kg dry matter). The values of M_e_ are relatively small comparing to those of M_(t)_ or M_o_. The error due to the simplification is negligible [46,47,48], thus the moisture ratio was calculated as follows: (2)$$MR=\frac{M_{\left(t\right)}}{M_{0}}$$

Table Curve 2D Windows v2.03 was used to fit the basic drying models to the measured MR determined accordingly to Equation (2). There are several drying models in the literature that can be used to describe the kinetics of drying plant materials. For drying model selection, drying curves were fitted to five well known thin drying models, including the modified Page model. Henderson–Pabis, logarithmic, Midilli-Kucuk, and Weibull ones. The best fit was determined using two parameters: the value of the coefficient of determination (R^2^) and root-mean squared error (RMSE). A model fits better if the value of R^2^ is closer to 1, and the RMSE value is closer to 0, using the following equations:(3)$$R^{2}=\frac{\sum_{i=1}^{N}\left(MR_{pre_{i}}-{\bar{MR}}_{\exp}\right)}{\sum_{i=1}^{N}\left(MR_{\exp_{i}}-{\bar{MR}}_{\exp}\right)}$$
(4)$$RMSE=\sqrt{\frac{1}{N}\cdot \sum_{i=1}^{N}{\left(MR_{\exp_{i}}-MR_{pre_{i}}\right)}^{2}}$$
where *MR* is moisture ratio, $\bar{MR}$ is the mean value of moisture ratio, “*pre*” and “exp” indicate predicted and experimental values, respectively, while “*i*“ indicates subsequent experimental data and *N* is the number of observations.

Tests conducted in this study proved that the best fitting was obtained for the modified Page model:(5)$$MR=A\exp\left(-k\tau^{n}\right)$$
where *A, n*, and *k* are constants.

### 3.4. Solid-Phase Micro Extraction (SPME) Analysis

HS-SPME analysis (30 min exposure to a 2 cm DVB/CAR/PDMS fiber, Supelco, Bellefonte, PA, USA, followed by analyte desorption at 220 °C for 3 min) was performed on Varian CP-3800/Saturn 2000 apparatus (Varian, Walnut Creek, CA, USA) equipped with a Zebron ZB-5 MSI (30 m × 0.25 mm × 0.25 µm) column (Phenomenex, Shim-Pol, Poland). About 0.100 g of fresh or 0.150 g of dried sample was put in to headspace vials and kept in laboratory water bath at 70 °C. 0.5 mg of 2-undecanone (Sigma Aldrich, Saint Louis, MO, USA) as an internal standard was added. 

### 3.5. GC-MS Analysis

The GC oven temperature was programmed from 50 °C, to 130 °C at rate 4.0°C. then to 180 °C at rate 10.0 °C, then to 280 °C at rate 20.0 °C. Scanning was performed from 35 to 550 *m*/*z* in electronic impact (EI) mode at 70 eV. Samples were injected at a 1:10 split ratio and helium gas was used as the carrier gas at a flow rate of 1.0 mL·min^−1^. Analyses were run in triplicate.

### 3.6. Hydrodistillation of Essential Oil (EO)

Hydrodistillation of EOs was carried out by applying a Deryng apparatus. About 200 g of fresh sample or 100 g of dried sample was placed in 2 L round bottom flask with 500 mL of added distilled water. Yield was assessed as a measured volume of essential oil.

### 3.7. Identification and Quantification of Volatile Compounds

Identification of all volatile constituents obtained by HS-SPME analysis and hydrodistillation were based on comparison of experimentally obtained compound mass spectra with mass spectra available in NIST14 database. Also the experimentally obtained retention indeces (RI) by Kovats were compared with RI available in the NIST WebBook and literature data [32]. The quantification analysis was performed using ACD/Spectrus Processor (Advanced Chemistry Development, Inc., Toronto, ON, Canada) through the integration of the peak area of the chromatograms. 

### 3.8. Statistical Analysis

The data from drying kinetics were subjected to the analysis of variance using Tukey’s test (*p* < 0.05) and the data from quantitative essential oil and volatile constituents were subjected to the analysis of variance using Duncan’s test (*p* < 0.05), all using the STATISTICA 13.3 software for Windows (StatSoft, KrakowPoland).

## 4. Conclusions

One hundred constituents were identified in the volatile profile of true lavender leaves, with *p*-cymen-8-ol (4.09% ± 0.67), a mixture of borneol and lavndulol (4.66% ± 0.69), *o*-cymene (4.81% ± 0.52), bornyl acetate (5.57% ± 0.82), (*E*)-caryophyllene (6.11% ± 1.48), eucalyptol (7.28% ± 1.06) and γ-cadinene (10.53±1.51) as a major ones. When various methods are applied during the drying process, this profile is strongly affected. The optimal drying method is dependent on the purpose of the product utilization. A most interesting fact is that the drying process may decrease the share of camphor, while increasing the share of linalool and linalyl acetate which are the most desirable in components in true lavender aroma. This result may be a good starting point for considering the improvement of the value of true lavender leaves in comparison to its flowers for flavoring applications.

## Figures and Tables

**Figure 1 molecules-24-00764-f001:**
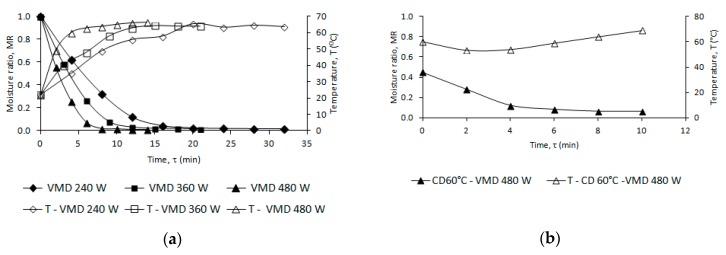
(**a**) Drying kinetics of true lavender leaves samples processed using VMD at magnetron powers 240, 360 and 480 W; (**b**) Drying kinetics of true lavender leaves samples processed using CD at temperatures of 50, 60 and 70 °C; (**c**) Drying kinetics of true lavender leaves samples processed using VMFD at 480 W after CPD at temperature 60 °C.

**Figure 2 molecules-24-00764-f002:**
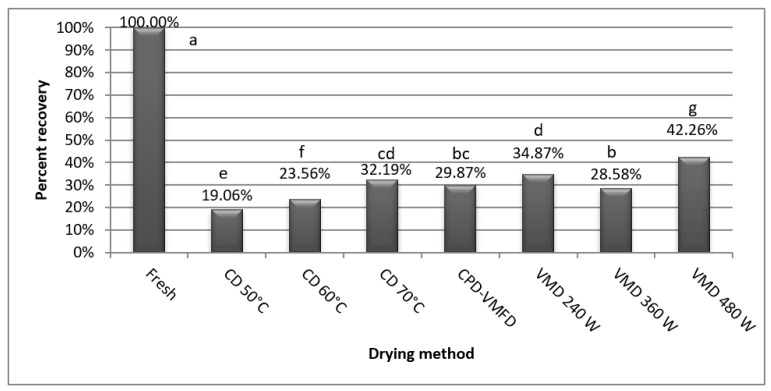
Percent recovery of essential oil of true lavender leaves after applying various drying methods.

**Table 1 molecules-24-00764-t001:** Final moisture content (M_fwb_), maximum temperature of the sample T, convective drying time (τ), vacuum microwave drying time (τ_1_), and constants A, k and n of the modified Page model describing the drying kinetics.

Drying Conditions	*A*	Constants *K*	*n*	R^2^	RMSE	τ	τ_1_	T (°C)	*M_fwb_ (%)*
CD 50 °C	1.000	0.0201	0.953	0.9984	0.0125	245	-	50	7.18
CD 60 °C	1.000	0.0125	1.173	0.9991	0.0104	145	-	60	7.09
CD 70 °C	1.000	0.0202	1.150	0.9983	0.0156	135	-	70	7.42
VMD 240 W	1.000	0.0736	1.328	0.9989	0.0127	-	32	64	6.78
VMD 360 W	1.000	0.1205	1.358	0.9991	0.0104	-	21	65	6.90
VMD 480 W	1.000	0.2339	1.300	0.9991	0.0111	-	14	66	6.87
**CPD 60°C + VMFD 480 W**	**0.449**	**0.2895**	**0.893**	**0.9982**	**0.0155**	**60**	**10**	**64**	**7.02**

**Table 2 molecules-24-00764-t002:** Volatile constituents of fresh true lavender leaves.

Compound	RT (min)	Retention Indeces (RI)			Content [%] ^4^
		RI_lit ^1^	RI_lit ^2^	RI_exp ^3^	
**1-Penten-3-ol**	2.407	-	684	686	Tr ^5^
**(*Z*)-3-Hexenal**	3.755	797	810	808	0.23 ± 0.14
**(*E*)-2-Hexenal**	4.765	846	854	857	0.33 ± 0.17
**(*Z*)-3-Hexen-1-ol**	4.821	850	857	859	1.75 ± 0.35
**1-Hexanol**	5.087	863	868	871	0.32 ± 0.09
**(*E,E*)-2,4-Hexadienal**	6.113	909	911	913	0.15 ± 0.09
**5.5-Dimethyl-1-vinylbicyclo[2.1.1]hexane**	6.380	-	921	924	tr
**Tricyclene**	6.479	921	926	928	0.17 ± 0.03
**α-Thujene**	6.591	924	930	932	0.12 ± 0.05
**α-Pinene**	6.788	932	939	940	0.30 ± 0.07
**Camphene**	7.209	946	954	955	0.92 ± 0.19
**3,7,7-Trimethyl-1.3.5-cycloheptatriene**	7.840	-	972	976	tr
**Sabinene**	7.911	696	976	978	0.11 ± 0.03
**1-Octen-3-ol**	8.038	974	979	982	0.72 ± 0.06
**3-Octanone**	8.260	979	986	988	0.22 ± 0.03
**β-Myrcene**	8.415	988	991	993	0.52 ± 0.22
**Mesitylene**	8.512	994	995	996	tr
**n-Decane**	8.681	1000	1000	1000	0.19 ± 0.04
**α-Phellandrene**	8.850	1002	1005	1007	0.49 ± 0.28
**3-Carene**	9.031	1008	1011	1013	1.60 ± 0.66
***m*-Cymene**	9.397	1020	1024	1026	2.58 ± 0.33
***p*-Cymene**	9.482	1022	1030	1028	4.81 ± 0.52
**Limonene**	9.634	1024	1030	1033	3.42 ± 1.16
**Eucalyptol**	9.692	1026	1031	1035	7.28 ± 1.06
**β-*cis*-Ocimene**	9.902	1032	1038	1042	0.16 ± 0.03
**β-*trans*-Ocimene**	10.240	1044	1050	1053	0.14 ± 0.04
**γ-Terpinene**	10.605	1054	1059	1063	0.11 ± 0.03
***trans*-Sabinene hydrate**	10.886	1065	1070	1071	0.23 ± 0.05
***cis*-Linalool oxide**	11.041	1067	1074	1076	0.13 ± 0.03
**unknown**	11.167	-	-	1079	tr
***m*-Cymenene**	11.419	1082	1085	1086	0.50 ± 0.04
***p*-Mentha-2.4(8)-diene**	11.519	1085	1088	1089	0.34 ± 0.10
***p*-Cymenene**	11.602	1089	1091	1091	0.30 ± 0.03
**Camphenone**	11.840	1095	1096	1097	0.26 ± 0.02
**Linalool**	11.953	1095	1096	1100	0.42 ± 0.03
**1.3.8-p-Menthatriene**	12.206	1108	1110	1108	0.10 ± 0.02
**1-Octen-3-ol acetate**	12.360	1110	1112	1114	3.80 ± 0.52
***cis*-*p*-Menth-2-en-1-ol**	12.556	1118	1121	1120	0.18 ± 0.04
***trans*-*p*-Mentha-2.8-dien-1-ol**	12.724	1119	1122	1125	0.64 ± 0.19
***cis*-*p*-Mentha-2.8-dien-1-ol**	13.173	1133	1137	1139	0.26 ± 0.03
***trans*-*p*-Menth-2-en-1-ol**	13.327	1136	1140	1144	0.49 ± 0.08
**Camphor**	13.496	1141	1146	1149	2.09 ± 0.29
**Tetrahydrolavandulol**	13.960	1157	1161	1162	0.48 ± 0.09
**Borneol + Lavandulol**	14.240	1165	1169	1170	4.66 ± 0.69
**Melilotal**	14.450	1179	1182	1176	tr
**Terpinen-4-ol**	14.631	1174	1177	1181	0.59 ± 0.07
***m*-Cymen-8-ol**	14.774	1176	1179	1184	2.09 ± 0.25
***p*-Cymen-8-ol**	14.914	1179	1182	1188	4.09 ± 0.67
**α-Terpineol**	15.082	1186	1189	1193	0.31 ± 0.06
**Myrtenol**	15.278	1194	1195	1198	0.20 ± 0.14
***trans*-Piperitol**	15.671	1207	1208	1210	0.65 ± 0.07
***cis*-Carveol**	16.035	1215	1217	1222	0.37 ± 0.07
**(*Z*)-Ocimenone**	16.159	1226	1229	1226	0.26 ± 0.07
***exo*-Fenchyl acetate**	16.356	1229	1232	1232	0.49 ± 0.04
***cis*-Verbenol**	16.623	1237	1244	1240	tr
**Cumin aldehyde**	16.748	1238	1241	1244	1.92 ± 0.59
**Carvone**	16.874	1246	1243	1247	1.08 ± 0.28
**Geraniol**	17.055	1249	1252	1253	0.33 ± 0.29
**Linalyl acetate**	17.263	1254	1257	1259	2.21 ± 0.73
**Geranial**	17.529	1264	1267	1267	0.10 ± 0.08
***trans-*Carvone oxide**	18.021	1273	1276	1281	0.33 ± 0.07
**Bornyl acetate**	18.301	1284	1285	1288	5.57 ± 0.82
**Lavandulyl acetate**	18.428	1288	1290	1292	1.72 ± 0.25
**Terpinen-4-ol acetate**	18.761	1299	1299	1301	0.18 ± 0.02
**unknown**	19.124	-	-	1314	0.61 ± 0.09
**Myrtenyl acetate**	19.435	1324	1326	1326	0.16 ± 0.06
**δ-Elemene**	19.749	1335	1337	1337	tr
**α-Terpinyl acetate**	20.036	1346	1349	1347	0.26 ± 0.08
**α-Cubebene**	20.179	1348	1351	1351	tr
**α-Longipinene**	20.351	1350	1352	1357	0.18 ± 0.01
**unknown**	20.465	-	-	1361	0.31 ± 0.05
**Silphiperfola-4.7(14)-diene**	20.578	1358	1362	1365	tr
**Neryl acetate**	20.748	1359	1364	1371	0.26 ± 0.06
**α-Copaene**	21.134	1374	1376	1383	0.14 ± 0.02
**Geranyl acetate**	21.248	1379	1381	1387	0.49 ± 0.11
**α-Bourbonene**	21.375	1387	1388	1391	tr
**unknown**	21.461	1394	1396	1394	tr
**β-Longipinene**	21.634	1400	1400	1399	0.26 ± 0.06
**Sesquithujene**	21.833	1405	1405	1409	tr
**α-Cedrene**	22.049	1410	1411	1420	1.01 ± 0.25
**(*E*)-Caryophyllene**	22.176	1417	1419	1427	6.11 ± 1.48
**α-Bergamotene**	22.506	1432	1435	1443	0.87 ± 0.33
**Cadina-3.5-diene**	22.745	-	1458	1455	1.12 ± 0.40
**(*E*)-β-Farnesene**	22.889	1454	1457	1462	1.35 ± 0.38
***cis*-Muurola-4(15).5-diene**	23.084	1465	1466	1472	1.44 ± 0.45
**4-*epi*-α-Acoradiene**	23.155	1474	1475	1475	0.19 ± 0.00
**Germacrene D**	23.441	1484	1481	1489	0.58 ± 0.17
**β-Himachalene**	23.629	1500	1500	1498	tr
**unknown**	23.741	1502	-	1505	tr
**α-Bulnesene**	23.840	1509	1509	1511	0.92 ± 0.16
**γ-Cadinene**	24.023	1513	1513	1523	10.53 ± 1.51
***cis*-Calamenene**	24.149	1528	1529	1531	0.65 ± 0.07
**10-*epi*-Cubebol**	24.290	1533	1535	1540	0.11 ± 0.05
**α-Cadinene**	24.402	1537	1538	1547	0.12 ± 0.02
**Cadala-1(10).3.8-triene**	24.473	-	1555	1552	tr
***trans*-Cadinene ether**	24.669	1557	-	1564	0.35 ± 0.09
**unknown**	24.851	-	-	1576	0.13 ± 0.05
**Spathulenol**	24.950	1577	1578	1582	0.25 ± 0.05
**Caryophyllene oxide**	25.158	1582	1583	1595	3.31 ± 0.18
**1-*epi*-Cubenol**	25.552	1627	1628	1628	0.56 ± 0.03
**τ-Cadinol**	25.860	1635	1340	1656	2.04 ± 0.55
**unknown**	26.056	-	-	1673	0.11 ± 0.03
**14-Hydroxy-4.5-dihydrocaryophyllene**	*26.407*	1706	1706	1706	0.21 ± 0.11
**unknown**	26.911	1760	1761	1764	0.23 ± 0.02

^1^ Retention indices according to Adams [32]; ^2^ Retention indices according to NIST14 database; ^3^ Relative retention indices calculated against *n*-alkanes; ^4^ % calculated from TIC data; ^5^ tr. < 0.1%.

**Table 3 molecules-24-00764-t003:** Variability of major volatile constituents, linalool and total essential oil of true lavender leaves caused by various drying methods.

Compound		Drying Method						
	Fresh	CD 50 °C	CD 60 °C	CD 70 °C	CPD-VMFD	VMD 240 W	VMD 360 W	VMD 480 W
	Content [%] ^1^							
***p*-Cymene**	2.58 ^a^	2.73 ^c^	2.72 ^c^	1.76 ^d^	2.01 ^de^	2.15 ^e^	2.27 ^e^	3.50 ^b^
***o*-Cymene**	4.81 ^a^	6.26 ^c^	5.65 ^d^	3.05 ^f^	3.69 ^e^	4.62 ^g^	4.73 ^g^	8.08 ^b^
**Limonene**	3.42 ^a^	3.27 ^f^	3.47 ^f^	1.31 ^e^	3.08 ^cf^	1.97 ^de^	2.41 ^cd^	6.99 ^b^
**Eucalyptol**	7.28 ^a^	5.01 ^bc^	3.71 ^d^	5.12 ^b^	3.98 ^cd^	3.25 ^d^	3.74 ^d^	3.44 ^d^
**1-Octen-3-ol acetate**	3.80 ^a^	2.70 ^de^	2.82 ^de^	4.23 ^c^	6.22 ^b^	2.10 ^e^	4.42 ^c^	3.68 ^cd^
**Camphor**	2.09 ^a^	2.32 ^b^	1.40 ^d^	1.89 ^c^	0.40 ^f^	1.12 ^e^	0.34 ^f^	0.39 ^f^
**Borneol + Lavandulol**	4.66 ^a^	7.63 ^b^	5.75 ^d^	6.07 ^d^	1.37 ^e^	4.78 ^c^	1.46 ^e^	1.35 ^e^
***m*-Cymen-8-ol**	2.09 ^a^	3.18 ^c^	2.48 ^d^	3.67 ^b^	0.02 ^e^	2.74 ^d^	0.08 ^e^	0.07 ^e^
***p*-Cymen-8-ol**	4.09 ^a^	6.31 ^c^	6.05 ^c^	7.17 ^b^	1.10 ^d^	6.07 ^c^	1.04 ^d^	0.89 ^d^
**Cumin aldehyde**	1.92 ^a^	3.59 ^c^	3.74 ^c^	4.48 ^b^	0.59 ^d^	4.30 ^b^	0.66 ^d^	0.59 ^d^
**Linalyl acetate**	2.21 ^a^	3.46 ^d^	11.06 ^b^	4.23 ^d^	1.60 ^e^	5.29 ^c^	1.66 ^e^	1.75 ^e^
**Bornyl acetate**	5.57 ^a^	3.54 ^c^	2.36 ^e^	4.04 ^b^	0.07 ^f^	3.07 ^d^	0.07 ^f^	0.14 ^f^
(***E***)-**Caryophyllene**	6.11 ^a^	2.11 ^d^	2.78 ^c^	3.51 ^f^	6.38 ^b^	4.85 ^e^	5.28 ^e^	3.98 ^f^
**γ-Cadinene**	10.53 ^a^	3.67 ^f^	4.43 ^ef^	4.74 ^e^	8.48 ^cd^	7.80 ^d^	9.20 ^c^	5.88 ^b^
**Caryophyllene oxide**	3.31 ^a^	2.43 ^c^	1.63 ^b^	2.12 ^bc^	2.24 ^bc^	2.11 ^bc^	2.47 ^c^	1.89 ^bc^
**τ-Cadinol**	2.04 ^a^	1.62 ^d^	1.44 ^d^	2.78 ^c^	2.74 ^c^	3.37 ^bc^	3.85 ^b^	1.74 ^d^
**Ʃ-Linalool** **TOTAL essential oil [mL 100g^−1^ dw] ^2^**	66.51 ^a^ 0.38 ^a^ 3.082 ^a^	59.83 ^b^ 4.32 ^c^ 0.588 ^e^	61.42 ^b^ 6.33 ^b^ 0.726 ^f^	60.17 ^b^ 4.71 ^c^ 0.992 ^cd^	43.97 ^c^ 0.76 ^ad^ 0.921 ^bc^	59.59 ^b^ 6.62 ^b^ 1.075 ^d^	43.68 ^c^ 0.73 ^ad^ 0.881 ^b^	42.47 ^c^ 1.16 ^d^ 1.302 ^g^

^1^ Values followed by the same letter within a row are not significantly different (*p* > 0.05, Duncan’s test); ^2^ Values obtained from steam distillation in Deryng apparatus.

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
