# Peer review of "HS-SPME Analysis of True Lavender (Lavandula angustifolia Mill.) Leaves Treated by Various Drying Methods"

_molecules, 2019, doi:10.3390/molecules24040764_

Round 1
Reviewer 1 Report
- grammar and style need to be improved
- probable mistakes in the chemical analysis:
m-cymene (RT 9.397 min.) before p-cymene (RT 9.482 min.)
cis-/trans-p-Mentha-2,8-dien-ol ... not rho
Cumin aldehyde not Cumaldehyde
beta-Farnesen instead of Farmesene
sure about alpha-Chamipinene, gamma-Patchoulene und (Z)-Lanceol? (not found in lavender oil yet)
- use no more decimal places than the precision of the measurements provides
Author Response
Dear Reviewer,
thank you for your comments. We found them as a highly valuable and improving the quality of our work. Please, find our response for particular comments below:
1.
m-cymene (RT 9.397 min.) before p-cymene (RT 9.482 min.)
cis-/trans-p-Mentha-2,8-dien-ol ... not rho
Cumin aldehyde not Cumaldehyde
beta-Farnesen instead of Farmesene
All corrected
2. sure about alpha-Chamipinene, gamma-Patchoulene und (Z)-Lanceol? (not found in lavender oil yet)
We had overviewed some articles related to lavender oil composition and indeed did not find the mentioned compounds. In this case, we had decided that it will be better to leave those compounds as “unknowns”. Their mass spectra are added to supplementary materials, in case that someone will be able to identified them.
3. use no more decimal places than the precision of the measurements provides
All decimal places are based on the measurement method precision.
Your sincerly,
Jacek Łyczko
Reviewer 2 Report
Some questions and recommendations are in pdf-file.

Author Response
Dear Reviewer,
Thank you for your comments and notes. In our opinion, they strongly improve the quality of our work. Please find below our responses for your comments. I skipped here the typos errors and some little changes. You may track them in the revised version of the manuscript.
1. Is it not strange to dry material with volatile compounds in vacuum? (line 23)
The results of the study on effects of vacuum level and microwave power on rosemary volatile composition during vacuum–microwave drying (Calin et al., 2011) revealed that an appropriate use of vacuum with microwave heating can be beneficial in terms of volatile composition, sensory properties and time of drying, which is significantly longer for convective drying under atmospheric pressure which expose dried material to oxidation at elevated temperature and thus chemical alterations leading to decreasing of bioactive potential expressed by the content of polyphenols and antioxidant activity (Wojdylo et al., 2014).
2. What is Page model? (line 80)
The modified Page model is an empirical model in the form of mathematical equation describing the effect of drying time on moisture ratio (MR) which represents the moisture content of the dried sample. The modified Page model A•exp(-k•n) was obtained by incorporation of constant A to Page model exp(-k•n). The constant A enables modeling of microwave finishing drying which starts when the value of MR is lower than 1. Dimensionless moisture ratio MR ranges from 0 to 1 despite of the initial moisture content of the raw material in order to allow comparison of drying behavior of different moist materials.
3. Cepe (line 80)
Should be “True Lavender leaves” instead of “Cepe”.
4. Line 99
Revised for: “Combined CPD and VMFD using 480 W, shortened the drying time of leaves almost 18 times (folds) compared to CD at 50°C.”
5. Line 107
Revised for: “siphoning of water strongly bound to the cellular structure of the dried material”
6. Aren't volatile constituents a part of essential oils? have they different location in the leaves? (line 135)
Yes, there are a part of essential oils. In this case revised for: “were identified as a main true lavender leaves volatile components”.
7. Was the second procedure really necessary? Why do we need to remove all the water from material? (line 220)
The convective pre-drying procedure (CPD) in the combined drying method (CPD-VMFD) took only 60 min in order to remove relatively large amounts of water at satisfactory efficiency which is high at the initial stage of convective drying. However, the moisture content of the pre-dried material is too high taking into account the microbial safety of preserved product. Therefore, vacuum microwave finish drying (VMFD) is necessary to reduce the moisture content of the product to the safe level. Vacuum-microwave drying is much more effective than convective drying at the final stage of drying.
8. Line 227
Revised for: “the different drying time intervals were applied in order to ensure a similar energy input between subsequent measurements regardless of the microwave power level.”
9. What are the conditions of equilibrium? Line 232
At equilibrium conditions any change in moisture content of the dried material is observed despite of the time of drying.
10. There is no explanation of the fourth MR. (line 246)
The section on determination of R2 and RMSE was corrected.
(see equation in attached)
where MR is moisture ratio, (MR) ̅ is the mean value of moisture ratio, “pre” and “exp” applies predicted and experimental values, respectively, while “i“ indicates subsequent experimental data and N is the number of observations.
11. How was proved that results of SPME corresponds to the real concentrations of volatile compounds in air? Was a degree of absorption equal for each volatile compounds? (line 262)
The analysis of essential oil obtained from lavender leaves was performed. We had checked if the amounts of compounds and the ratio between them is corresponding to SPME results.
Moreover, the internal standard added to each SPME analysis was the determinant of the analysis repeatability.
Your sincerely,
Jacek Łyczko
